# MetaFunPrimer: an Environment-Specific, High-Throughput Primer Design Tool for Improved Quantification of Target Genes

Jia Liu,[a,f,g] Paul Villanueva,[a,f] Jinlyung Choi,[a,g] Santosh Gunturu,[c,h] Yang Ouyang,[b,h] Lisa K. Tiemann,[b,h] James R Cole,[b,c,h] Kathryn R. Glanville,[a,g] Steven J. Hall,[d] Marshall D. McDaniel,[e,g] Jaejin Lee,[a,g] ⓘ Adina Howe[a,f,g]

[a]Department of Agricultural and Biosystems Engineering, Iowa State University, Ames, Iowa, USA
[b]Department of Plant, Soil, and Microbial Sciences, Michigan State University, East Lansing, Michigan, USA
[c]Center for Microbial Ecology, Michigan State University, East Lansing, Michigan, USA
[d]Department of Ecology, Evolution, and Organismal Biology, Iowa State University, Ames, Iowa, USA
[e]Department of Agronomy, Iowa State University, Ames, Iowa, USA
[f]Bioinformatics and Computational Biology Department, Iowa State University, Ames, Iowa, USA
[g]Center for Advanced Bioenergy and Bioproducts Innovation, Urbana, Illinois, USA
[h]DOE Great Lakes Bioenergy Research Center, University of Wisconsin—Madison, Madison, Wisconsin, USA

Jia Liu and Paul Villanueva contributed equally to this work. Author order was determined by amount of leading contribution.

**ABSTRACT**   Genes belonging to the same functional group may include numerous and variable gene sequences, making characterizing and quantifying difficult. Therefore, high-throughput design tools are needed to simultaneously create primers for improved quantification of target genes. We developed MetaFunPrimer, a bioinformatic pipeline, to design primers for numerous genes of interest. This tool also enables gene target prioritization based on ranking the presence of genes in user-defined references, such as environment-specific metagenomes. Given inputs of protein and nucleotide sequences for gene targets of interest and an accompanying set of reference metagenomes or genomes, MetaFunPrimer generates primers for ranked genes of interest. To demonstrate the usage and benefits of MetaFunPrimer, a total of 78 primer pairs were designed to target observed ammonia monooxygenase subunit A (*amo*A) genes of ammonia-oxidizing bacteria (AOB) in 1,550 publicly available soil metagenomes. We demonstrate computationally that these *amo*A-AOB primers can cover 94% of the *amo*A-AOB genes observed in the 1,550 soil metagenomes compared with a 49% estimated coverage by previously published primers. Finally, we verified the utility of these primer sets in incubation experiments that used long-term nitrogen fertilized or unfertilized soils. High-throughput quantitative PCR (qPCR) results and statistical analyses showed significant differences in relative quantification patterns between the two soils, and subsequent absolute quantifications also confirmed that target genes enumerated by six selected primer pairs were significantly more abundant in the nitrogen-fertilized soils. This new tool gives microbial ecologists a new approach to assess functional gene abundance and related microbial community dynamics quickly and affordably.

**IMPORTANCE** Amplification-based gene characterization allows for sensitive and specific quantification of functional genes. There is often a large diversity of genes represented for functional gene groups, and multiple primers may be necessary to target associated genes. Current primer design tools are limited to designing primers for only a few genes of interest. MetaFunPrimer allows for high-throughput primer design for various genes of interest and also allows for ranking gene targets by their presence and abundance in environmental data sets. Primers designed by this tool improve the characterization and quantification of functional genes in

**Ad Hoc Peer Reviewers** ⓘ Robert Sanford, University of Illinois at Urbana Champaign; Sul Woo Jul, Chung-Ang University

Address correspondence to Adina Howe, adina@iastate.edu.

MetaFunPrimer allows for high-throughput primer design for various genes of interest and also allows for ranking gene targets by their presence and abundance in environmental datasets.

broad gene amplification platforms and can be powerful with high-throughput qPCR approaches.

**KEYWORDS** primer design, environment-specific, quantitative PCR, functional gene quantification, nitrification

Diverse microbes in our surrounding environments are key drivers of nutrient cycling and energy conversion necessary for our lives (1–3). To understand the role of these microbes in the environment, we characterize their community composition and structure, their diversity, and their function under various conditions. Efforts for characterizing microbiomes have been aided by the development of molecular techniques to amplify genes of interest in combination with their subsequent sequencing. Specifically, 16S rRNA gene amplicon sequencing has enabled high-throughput characterization of taxa or gene composition to inform community structure (4, 5). These methods are often limited to characterizing phylogenetic markers within a community and are not optimized for functional genes within microbial communities.

To characterize the functional potential of microbes, several approaches can be used. One method is to isolate and enrich representatives of a function of interest to identify and characterize functional traits and their hosts (6, 7). A challenge to this approach is that environmental microbes cultivated in the laboratory may not represent the microbes under actual environmental conditions (8–11). To complement the cultivation of isolates, culture-independent, sequencing-based approaches have been used to characterize functional genes (12–14). Specifically, metagenome sequencing of environmental DNA can be used to characterize diverse functional genes in environmental samples. However, it is often the case that these genes make up only a small fraction of the environmental DNA, which can result in a high cost to obtain desired but insufficient information of targeted functional genes (15). Another method to characterize functional genes is to target amplicons with PCR-based methods. Like 16S rRNA gene sequencing, these methods amplify certain target genes of interest. All amplicon-based approaches rely on the ability of primer sets to amplify these genes of interest. These primer sets and their subsequent amplifications are most effective if they are both sensitive and specific to target genes. Many existing primers have been developed based on gene sequences or genomes (16–19). In recent years, the increasing availability of metagenome sequencing has provided new opportunities for expanding or redesigning primers for target genes, especially for microbes that may not be cultivatable or have genomes available (20).

To capture as diverse a range of genes as possible, universal or degenerate primers have often been designed to quantify genes of interest. However, these primer pairs typically provide poor characterization at high specificity, especially when the target regions are short (21). An example of this limitation is community composition analysis through 16S rRNA amplicon sequencing using degenerate (universal) primer pairs for short variable regions. These primers are often limited to identifying microbes at the phylum, class, or order level and may not be reliable for identifying bacterial species or strains (22). Functional gene quantification using a degenerate primer set can also result in similar constraints, although an advantage to their usage is their ability to detect the presence of broad genes within a single assay.

PCR-based characterization of functional gene targets has recently been expanded with the development of high-throughput qPCR (HT-qPCR) platforms that can process thousands of PCRs in a single run and allows for numerous primer pairs and associated gene targets on a single run. In recent studies, hundreds of primer sets combined with HT-qPCR have been used simultaneously to characterize antibiotic resistance genes in environmental samples (23, 24). The emergence of HT-qPCR platforms increases the scale of PCR-based assays for functional genes of interest. Combined with novel gene information gained from metagenomes and new reference genomes, we can enhance our ability to characterize diverse functional genes in the environment; however, leveraging this technology is limited by a lack of software that allows users to design environment-specific primers for specific functional genes.

To address this need, we developed MetaFunPrimer, a pipeline that performs high-throughput primer design for targeting genes of interest identified in metagenome samples. This tool builds upon existing primer design software for developing PCR or qPCR primers, such as Primer3 (25), which can design primers for specific amplification conditions and product length outputs but are limited to a small number of primers and gene targets. MetaFunPrimer designs primers for targeted environmental-specific functional genes, evaluates these primers against hundreds of environmentally abundant functional genes, and characterizes the number of primer pairs that is required to capture the diversity in a given reference gene set.

Here, we demonstrate the use of MetaFunPrimer by designing primer pairs to target ammonia monooxygenase subunit A gene of ammonia-oxidizing bacteria (*amo*A-AOB) observed in soil as a specific target gene of interest. *amo*A-AOB genes were chosen as targets for functional primer design due to their important role in nitrogen cycling. *amo*A genes encode ammonia monooxygenase, an enzyme that is the main catalyst in ammonia oxidation. Ammonia oxidation is the first and rate-limiting step of the nitrification pathway which converts ammonia to nitrite and then to nitrate, the chemical form of nitrogen that may easily be lost from soils via leaching (26, 27). Generally, AOB species belong to either *Betaproteobacteria* or *Gammaproteobacteria*, which are subclasses of the class *Proteobacteria*, with the majority of AOB associated with genera *Nitrosococcus*, *Nitrosomonas*, and *Nitrosospira* (28, 29). *amo*A genes have been used previously as functional markers for analyzing AOB diversity (16, 30, 31), and several primer pairs for conserved regions of *amo*A-AOB genes have been used previously for studying its function (16–19).

In this study, we evaluate the diversity of *amo*A-AOB genes observed in 1,550 publicly available soil metagenomes, evaluate the sensitivity and specificity of previously published primers to detect these genes, and use MetaFunPrimer to design primers for available *amo*A-AOB gene sequences identified in these 1,550 soil metagenomes. To test the efficacy of the designed primers, we used them to characterize the *amo*A-AOB communities in a long-term agricultural experiment by crossing two crops (*Zea mays* L. and *Miscanthus* × *giganteus* Greef et Deu.) with two nitrogen (N) fertilizer rates (0 and 336 kg N ha$^{-1}$ y$^{-1}$). While this study focuses on *amo*A-AOB as a specific target gene of interest, MetaFunPrimer is broadly applicable to various genes of interest. An online tutorial of the use of MetaFunPrimer is available online at https://metafunprimer .readthedocs.io/en/latest/Tutorial.html.

## RESULTS

The primer design steps for *amo*A-AOB genes using MetaFunPrimer include the following: (i) characterization of reference *amo*A-AOB genes, (ii) weighting of target genes based on their presence and absence in soil metagenomes, (iii) design of primers for selected genes, and (iv) computational primer evaluation through alignment to target genes (Fig. 1, Table 1).

**Characterization of reference *amo*A-AOB genes.** A curated set of functional genes for *amo*A-AOB was obtained from the Ribosomal Database Project Fungene (version 9.6) (32). The set included amino acid sequences, nucleotide sequences, and their corresponding NCBI accession numbers for a total of 1,205 *amo*A-AOB genes. We aimed to design a minimal set of primer pairs to detect as many target genes as possible based on the presence of these genes in given metagenomes. Thus, the first step was to rank reference genes through alignment against metagenome sequences. Representative sequences of reference genes were first selected by clustering sequences based on their amino acid sequence similarity. Among the 1,205 *amo*A-AOB protein sequences, many sequences were observed to have a high degree of similarity. When sequences were clustered from 80% to 100% amino acid similarity, we found that clustering sequences at greater than 96% similarity resulted in the largest increase in the resulting total unique clusters (Fig. 2). The choice of clustering by similarity percentage influences the number of primer pairs, assays, and diversity that can be captured for primer design. Users can vary their selection

 

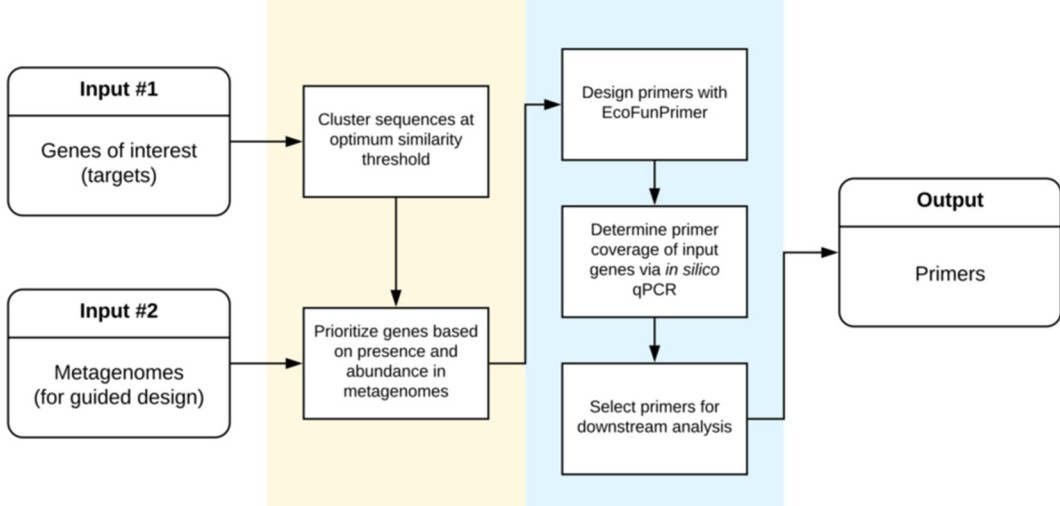

**FIG 1** Overview summarizing the MetaFunPrimer pipeline for gene primer design guided by inputs of reference genes and metagenomes.

of the appropriate number of clusters based on their goals and resources. In our case, we chose clustering at 96% similarity, which resulted in a total of 60 clusters. We found that the representative sequences from each cluster covered a wide diversity of *amo*A-AOB, including the genera *Nitrosomonas*, *Nitrosococcus*, and *Nitrosospira* (see Fig. S1 in the supplemental material).

**Weighting target genes based on soil metagenomes.** The representative protein sequences from each cluster were aligned to 1,550 publicly available soil metagenomes (see Table S1 in the supplemental material), with high alignment defined as having 97% sequence identity over the length of the reference gene. Each *amo*A-AOB associated gene identified in soil metagenomes was then ranked based on two criteria, as follows: estimated gene abundance (the total number of observations of each gene within all the metagenome sequences) and prevalence (the number of unique metagenomes where the gene was observed) (see Table S2 in the supplemental material). The abundance and prevalence of each representative gene were then normalized separately before taking their mean value to calculate each representative sequence's representation score (R-score). In our case, the clusters represented by the 10 representative sequences with the highest R-score accounted for a total of 720 *amo*A-AOB genes, comprising a total of 87% of the cumulative abundance of these genes observed in the soil metagenomes (Fig. 3).

**Design of primers for selected genes.** Given their representation in the soil metagenomes, we selected the nucleotide sequences of these 720 genes for further primer design. Embedded in MetaFunPrimer is EcoFunPrimer, which was developed by the Ribosomal Database Project (RDP) at Michigan State University (https://github.com/rdpstaff/EcoFunPrimer). EcoFunPrimer is a primer design tool which is capable of generating degenerate or nondegenerate primers based on input genes and parameters defined by the user. Using the 720 genes of interested identified above and allowing

**TABLE 1** Data associated with MetaFunPrimer in the design of soil-abundant *amo*A-AOB genes

| Data associated with MetaFunPrimer | Type | Results for this study |
|---|---|---|
| Curated *amo*A-AOB genes from functional gene database | Input | 1,205 nucleotide and amino acid sequences |
| Soil metagenomes | Input | 1,550 soil metagenomes |
| Optimal clustering similarity found (recommended by MetaFunPrimer) (%) | Parameter | 96 |
| Gene clusters included (recommended by MetaFunPrimer) | Parameter | 10 gene clusters |
| Prioritized genes based on input 1 and 2; total no. of soil abundant *amo*A-AOB genes | Output | 720 genes |
| Nondegenerate primers | Output | 78 primer pairs |
| Total no. (%) of soil-abundant *amo*A-AOB genes targeted by final primer set | Output | 676 (93.89) |

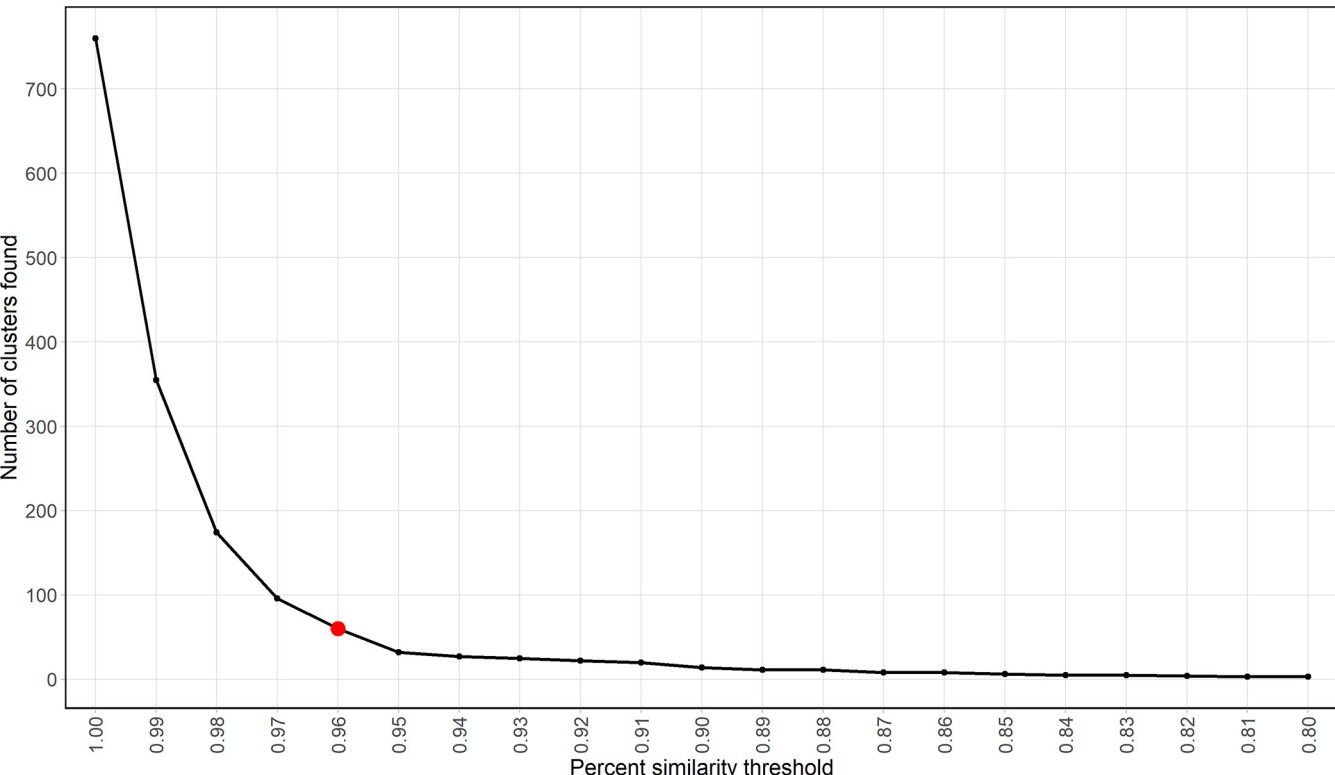

**FIG 2** Number of *amo*A-AOB clusters found by CD-HIT over a range of percent similarity thresholds. A total of 60 clusters were selected based on 96% amino acid similarity of *amo*A-AOB genes (indicated by red point). Clusters were found using CD-HIT with a word size of 5 for each of the similarity thresholds indicated, and the 96% threshold was chosen based on first-order difference calculation.

at most 6 degenerate forms for each primer resulted in 20 degenerate and 8 nondegenerate primer pairs targeting *amo*A-AOB genes (see Table S3 in the supplemental material). MetaFunPrimer converted these 28 primer pairs into their nondegenerate forms, resulting in 181 single nondegenerate primer pairs and evaluated their amplification sequentially through *in silico* PCR against the 720 targeted reference genes. In some cases, EcoFunPrimer may generate multiple primer pairs for the same gene target; thus, our software optimizes and selects the minimal set of primer pairs that can exclusively target the maximal diversity of functional genes of interest. For our gene targets, this resulted in a final set of 78 nondegenerate primer pairs for this primer design (see Table S4 in the supplemental material). Overall, the resulting primer pairs were predicted to *in silico* amplify a total of 676 of the targeted 720 soil-abundant *amo*A-AOB genes observed from soil metagenomes.

Finally, to compare our designed primers to preexisting ones (see Table S5 in the supplemental material), we summarized previously published *amo*A-AOB primers (16–19) to single nondegenerate forms without ambiguity. The MetaFunPrimer *in silico* amplification procedure was performed using these primer pairs to evaluate their alignment to the 720 soil-abundant *amo*A-AOB genes. In total, 49% (356/720) of these genes would be detected using pre-existing primer pairs, while the primers designed by MetaFunPrimer resulted in 94% (676/720) detection (Table 2). The computational analysis showed that primers designed using MetaFunPrimer tend to have higher coverage than pre-existing primers within each of the 10 soil-abundant *amo*A-AOB clusters (Table 2). In cluster 3, for instance, 96% (273/285) of the *amo*A-AOB genes would be targeted by primers designed using MetaFunPrimer, while pre-existing primers would amplify only 19% (55/285) of these genes (Table 2).

**Experimental validation.** In order to validate the resulting 78 *amo*A-AOB primer pairs from MetaFunPrimer, we used 96 soil DNA samples obtained from an incubation experiment using agricultural soils from a long-term cropping system experiment that

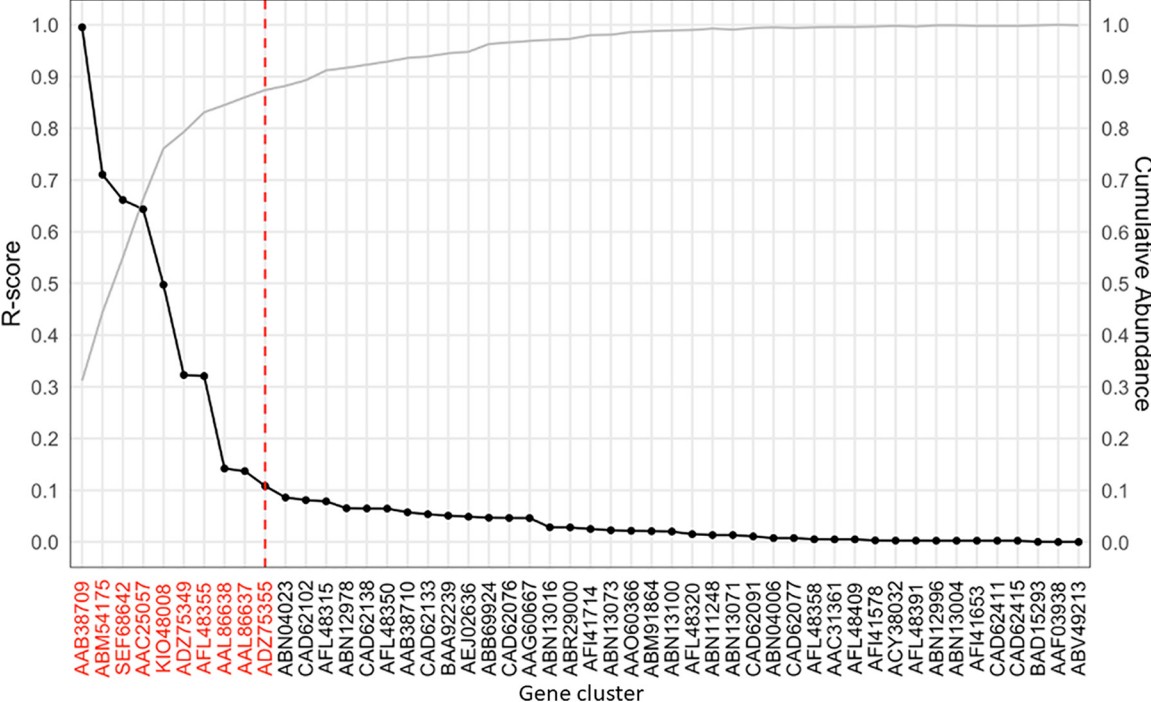

**FIG 3** Known *amo*A-AOB genes ranked by representation score (R-score, the mean of the scaled abundance and prevalence) and the estimated cumulative abundance of each gene in 1,550 soil metagenomes. The protein sequences in red indicate those *amo*A-AOB gene clusters and their associated genes that were selected for primer design based on the cumulative R-score in reference metagenomes.

had different nitrogen fertilizer treatment levels (0 and 336 kg N ha$^{-1}$ y$^{-1}$). In total, we found the targets of 70 primer pairs were amplified in at least 3 soil samples, and 93 samples had both 16S rRNA gene amplification and multiple amplifications by *amo*A-AOB primer pairs. Nonmetric multidimensional scaling (NMDS) based on Bray-Curtis distances (stress, 0.048) and analysis of similarities (ANOSIM) ($R = 0.66$; $P < 0.001$) showed that amplifications of *amo*A-AOB genes from the two fertilizer rate groups were significantly different from one other (Fig. 4).

Based on the differences between the two fertilizer rate groups, six primer pairs were selected for further absolute quantifications with comparisons to synthetic standards for each targeted gene (see Table S6 in the supplemental material). The absolute gene copy numbers measured for each of the six primer pairs were significantly greater in soil receiving N fertilizer for 5 years ($t = -4.69$, $P < 0.001$) (Fig. 5). The organisms associated with the amplified target genes are *Nitrosospira* sp. Wyke8 (amoA_AOB_p35) and *Nitrosolobus multiformis* (amoA_AOB_p31, amoA_AOB_p42, and amoA_AOB_p45).

**TABLE 2** Comparison of *amo*A-AOB primers in the literature[a] to those designed in this study

| Soil-abundant amoA-AOB cluster | Gene representative | No. of gene sequences within each cluster | Targeting rate[b] of | |
| --- | --- | --- | --- | --- |
| | | | Previously published primers | MetaFunPrimer primers |
| 1 | AAB38709 | 20 | 3 (15.00) | 19 (95.00) |
| 3 | SEF68642 | 285 | 55 (19.30) | 273 (95.79) |
| 4 | KIO48008 | 320 | 255 (79.69) | 304 (95.00) |
| 5 | AAC25057 | 65 | 30 (46.15) | 53 (81.54) |
| 6 | AAL86637 | 5 | | 3 (60.00) |
| 7 | AAL86638 | 11 | 10 (90.91) | 11 (100.00) |
| 28 | ABM54175 | 2 | | 2 (100.00) |
| 29 | ADZ75349 | 8 | 3 (37.50) | 7 (87.50) |
| 52 | AFL48355 | 2 | | 2 (100.00) |
| 58 | ADZ75355 | 2 | | 2 (100.00) |
| Total | | 720 | 356 (49.44) | 676 (93.89) |

[a]Detailed information of previously published primer pairs can be found in Table S5.
[b]Target rate is the number of genes within the associated cluster that can be amplified by given primer sets divided by total number of genes in the cluster (%).

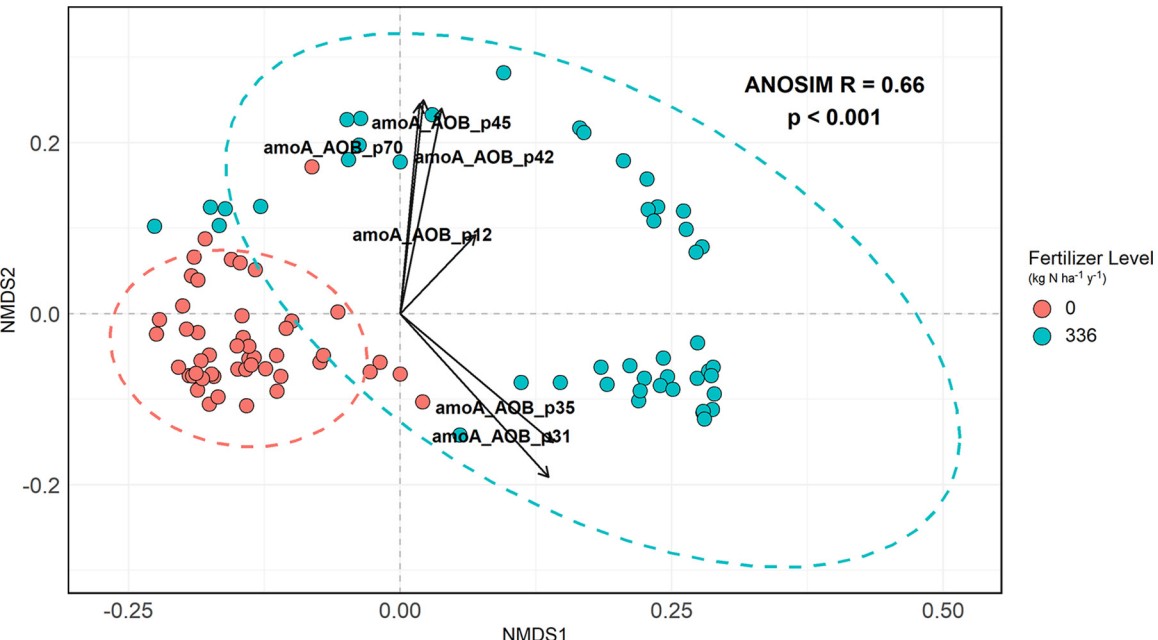

**FIG 4** Nonmetric dimensional scaling ordination (NMDS) plot of $C_T$ values from 78 primer pairs designed by MetaFunPrimer for amplifying *amo*A-AOB DNA in 96 soil samples. Stress was 0.048. The color of the points indicates the fertilizer treatment each sample received (red for 0 kg N ha$^{-1}$ y$^{-1}$; blue for 336 kg N ha$^{-1}$ y$^{-1}$), and the dashed ellipses represent the 95% confidence intervals for each treatment group. The arrows indicate the factor loading of 6 primer pairs chosen for further analysis as described in the text.

## DISCUSSION

Amplicon-based approaches for characterizing functional genes provide an approach that strongly complements metagenome sequencing. In comparison to metagenome sequencing, HT-qPCR approaches have the potential to be more affordable and informative due to the targeted amplification of genes of interest and can be used for standardized surveys of microbial communities and their functions (33). The opportunities of HT-qPCR approaches and amplicon-based approaches depend strongly on the reliability of

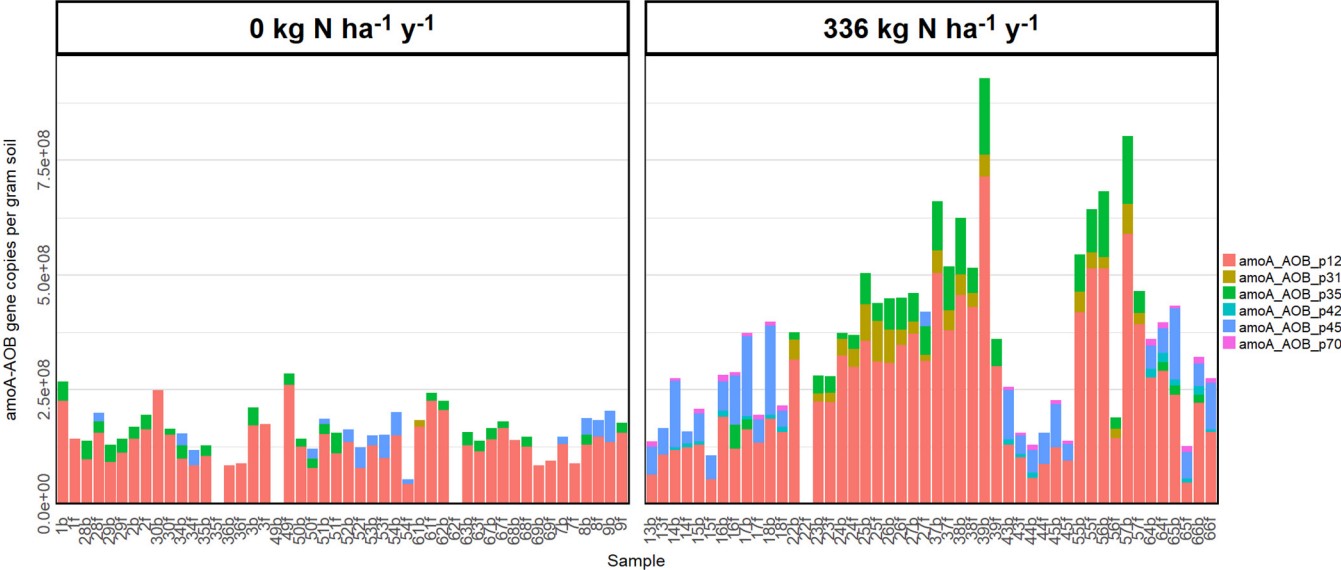

**FIG 5** Stacked bar plot of gene copy numbers amplified by the 6 *amo*A-AOB primer pairs identified for further analysis. HT-qPCR was used to quantify the gene copy number amplified by each of these 6 primer pairs in 96 soil samples across 2 different levels of fertilization treatment. The abundance of genes in the fertilized samples was significantly higher than that in the unfertilized samples (t = −4.69, P < 0.001).

primer design to target genes of interest (34). We introduce the MetaFunPrimer pipeline for high-throughput primer design and demonstrate its effectiveness by capturing a broad diversity of relevant genes associated with ammonia oxidation within soil metagenomes in combination with HT-qPCR.

Nitrogen cycling genes are one of the most challenging targets for amplicon approaches as they are carried by highly diverse microorganisms, including heterotrophic nitrifying microorganisms, denitrifying bacteria, anammox bacteria, nitrifying archaea, and denitrifying fungi (35). Previously, there have been numerous efforts to design primers for *amo*A and other nitrogen cycling genes (e.g., Table S5), but existing primers detect a limited range of the phylogenetically diverse genes and often result in misinterpretation (36). Our analysis supports these previous observations that currently existing primers capture less than one-half of the *amo*A-AOB genes in soil metagenomes. Using MetaFunPrimer, we developed 78 nondegenerate primer pairs to improve the quantification of these genes in soil metagenomes and increase the *in silico* detection of *amo*A-AOB genes from 49% coverage up to 94% coverage of observed genes in 1,550 soil metagenomes. Moreover, we demonstrate the usage of our designed primers in conjunction with HT-qPCR to detect the potential contributions of specific organisms to ammonia oxidation. By utilizing metagenome-derived sequences for primer design, this approach can expand our ability to detect uncharacterized bacterial species that play important roles in ammonia oxidation in the environment. Notably, in the 1,550 soil metagenomes used in this study, *amo*A-AOB genes account for less than 0.002% of reads in metagenome libraries and thus comprise only a fraction of each generated metagenome. In such cases where a gene of interest has a very low relative abundance, HT-qPCR approaches or even a conventional qPCR approach combined with MetaFunPrimer would allow a more sensitive detection of target genes.

In our *amo*A-AOB example, we aimed for a reasonable number of primer pairs (e.g., less than 100 primer pairs) to capture as many *amo*A-AOB genes in soils as possible. Generally, however, MetaFunPrimer inputs can be used to design primers for any user-inputted number of sequences, and this number could be varied to suit experimental capabilities or user-specific aims. Notably, some functional genes with higher sequence variability may require more primers than others. In cases where high sequence diversity exists, MetaFunPrimer will likely design numerous primer pairs, which may require numerous runs, even on HT-qPCR platforms. We have included various user options, such as the metagenome R-score, to help users rank and select the most relevant primer targets.

Another important attribute of MetaFunPrimer is the ability to rank target genes of primer design based on their presence in metagenomes. This feature allows for the selection of the most relevant genes based on previous observations of abundance and prevalence in reference metagenomes. Additionally, the selection of metagenomes as a reference for selecting primers can also be varied. For example, one could use inputs of metagenomes from only bioenergy-associated soils to prioritize microbial communities within specific agricultural sites. Alternately, genomes could be used as a reference for primer design, allowing users to weight primers for genes from known representatives.

Overall, we developed the MetaFunPrimer pipeline as a high-throughput primer design software and demonstrated its usage combined with HT-qPCR. This tool is appropriate for any targeted amplification platforms where primer design for specific genes of interests can be guided by available data sets, as we demonstrated in a recent paper which designed primers with the same approach and successfully measured microcystin-producing genes in hundreds of lake water samples (37). Within MetaFunPrimer, we also make available workflows for *in silico* comparisons of primers and gene targets. Similar to any primer design effort, further experimental validation is required, but computational efforts can help determine which candidates to test experimentally.

## MATERIALS AND METHODS

MetaFunPrimer takes as inputs the nucleotide and protein sequences of genes of interest, a file containing the mapping between a gene's nucleotide and protein sequence, and gene sequences for prioritization (such as metagenomes). The output of the pipeline is a set of primers that can be used to amplify selected functional genes. The major steps of MetaFunPrimer are to filter and rank genes of interest based on both diversity and representation in inputs and then to design and evaluate primer sequences for genes of interest (Fig. 1).

**Identifying environmentally representative gene clusters and determine target genes.** The first step in the MetaFunPrimer pipeline is to cluster input protein sequences over a range of amino acid sequence similarity thresholds in order to determine an optimal or user-defined similarity threshold. Specifically, CD-HIT (38, 39) is used to cluster sequences in the range of 80% to 100% (with 1% increments) similarity to determine the number of clusters found at each threshold. MetaFunPrimer will recommend a similarity threshold that optimizes the first-order difference, a criterion based on the symmetric derivative (37) (https://github.com/jialiu232/MetaFunPrimer_paper_info.git). However, users can select the most appropriate cluster similarity threshold based on their needs.

Next, MetaFunPrimer evaluates the presence of these genes in user-input reference sequences, i.e., metagenomes. For each cluster, the representative protein sequence (identified by CD-HIT) is aligned to reference sequences using DIAMOND (version 0.9.14) (40, 41). Each representative protein sequence is then ranked based on their R-score in reference sequences (i.e., in our case study, soil metagenomes). The R-score is defined as the mean of that gene's normalized abundance and prevalence among reference sequences. The representative genes for each cluster of sequences are ranked subsequently based on R-score in descending order, and gene clusters are included until the user-input threshold of the cumulative R-score (i.e., 80% in the case study) is reached. Genes that are associated with selected ranked clusters are considered genes of interest and consequently target genes for primer design and are converted into their corresponding nucleotide sequences.

**Designing and evaluating primers for genes of interest.** MetaFunPrimer uses selected gene sequences and user-defined parameters, such as amplicon product length and melting temperature ranges, for the subsequent primer design process (see Table S7 in the supplemental material). Within MetaFunPrimer, EcoFunPrimer is the primary tool used to design thermodynamically stable primer pairs from aligned nucleotide sequences. Depending on user-defined inputs, it is possible for primer outputs from EcoFunPrimer to have multiple degenerate forms. To evaluate primer effectiveness, MetaFunPrimer converts all primer outputs to nondegenerate forms (e.g., all possible primer pairs) of forward and reverse primers. Next, all primer pairs are evaluated via *in silico* PCR against the original set of reference genes provided by the user. A pair of primers was considered to amplify a gene product successfully if both the forward and reverse primers achieve a 100% match against a sequence. In some cases, a single reference gene may be targeted by multiple pairs of primers, and each primer pair can potentially target more than one gene. Thus, as a final step, MetaFunPrimer outputs the minimal number of primer sets to capture the maximum number of reference gene products (https://github.com/jialiu232/MetaFunPrimer_paper_info.git).

**Sample collection and high-throughput quantification of *amo*A-AOB genes.** Soil samples to test the 78 designed *amo*A-AOB primer pairs were from the Long-term Assessment of Miscanthus Productivity and Sustainability (LAMPS) site in Boone, IA (42°00′N, 97°44′W) (42). The 50-year mean annual temperature and precipitation from the site is 9.5°C and 895 mm, respectively. The soils are predominately Webster clay loams (Typic Endoaquolls), Canisteo clay loam (Typic Endoaquolls), and Clarion loam (Typic Hapludolls); this information was accessed from Web Soil Survey (http://websoilsurvey.sc.egov.usda.gov/; accessed 22 July 2021). Surface soil (0 to 10 cm) is characterized as having a wide range in pH (5.3 to 7.8, 1:1 deionized [DI] water), above-average cation exchange capacity (32.2 $\pm$ 1.9 cmol kg$^{-1}$), and soil organic matter (6.9% $\pm$ 2.8%) (43).

The LAMPS field experiment was established in 2015 and is arranged in a split-plot, randomized block design with four replications exposed to various experimental factors, including crop (maize versus Miscanthus) and urea nitrogen fertilizer rates. In the manuscript, we analyzed the effect of urea nitrogen fertilizer rates (0 and 336 kg N ha$^{-1}$ y$^{-1}$) on the AOB community. Composite soils were collected from four field replicates with a hammer core to a depth of 10 cm on 10 December to 11 December 2020. All field soil samples were divided into three additional treatment groups that had one of the following three treatments: (i) no additional treatment as control, (ii) 400 mg C kg$^{-1}$ in the form of glucose, and (iii) 60 mg N kg$^{-1}$ in the form of ammonium nitrate. Mesocosms were kept at room temperature (~23°C) and moisture (65% water-filled pore space [WFPS]) and vented once a week.

Surface soils were subsampled on days 5 and 87 of the incubation for DNA extraction. For 96 soil samples from the incubations (i.e., 2 crops × 2 fertilizer rates × 4 replicates × 3 additional treatments × 2 time points), genomic DNA was extracted using the epMotion 5075 liquid handler (Eppendorf, USA) following the protocol of MagAttract PowerSoil DNA EP kit (Qiagen, USA). The validation of 78 *amo*A-AOB primer pairs consisted of two steps using the Biomark HD high-throughput qPCR system (Fluidigm, USA). First, we observed the overall pattern of amplification through relative abundance calculated using the threshold cycle ($C_T$) values of all target genes. Next, we obtained absolute quantification of target genes using selected primer pairs. For the screening run containing 78 primer pairs and 96 samples, a single Biomark 96.96 dynamic array integrated fluidic circuit (IFC) (Fluidigm) was used, and the relative abundance was calculated using the $C_T$ values of the genes of interest and the reference gene (i.e., 16S rRNA genes). The 967F (5′-CAACGCGAAGAACCTTACC-3′) and 1194R (5′-ACGTCATCCCCACCTTCC-3′) primer pair was used to amplify 16S rRNA genes (44). The assay conditions, including primer concentration, can be found in Text S1 in the supplemental material. Four Biomark Flex Six gene expression IFCs were used to get the absolute quantification of target genes. Each Flex Six IFC contained 4 primer pairs in triplicate,

48 soil DNA samples, and 24 standards (i.e., 6 dilutions for each primer pair). Ten-fold-diluted soil DNA samples were used as the templates without preamplification. To prepare standard DNA templates for the Flex Six runs, each target gene fragment (i.e., linear, 500-bp gene fragments, including target region) was synthesized through Integrated DNA Technologies (IA, USA). The synthesized DNA concentration was adjusted to 0.1 ng/ml, and 10-fold serial dilutions were prepared ranging from $1.62 \times 10^1$ to $1.62 \times 10^6$ gene copies per reaction. Standard curves were obtained from the Flex Six runs (see Fig. S2 in the supplemental material). The limit of detection for all assays was $5.46 \times 10^6$ gene copies per gram of soil. According to the manufacturer's standard protocol, an IFC Controller HX (Fluidigm) was used for priming and loading the IFC. The default operating conditions were modified to 40 cycles of a 3-step PCR protocol (95°C for 15 s, 60°C for 30 s, and 72°C for 30 s) instead of 30 cycles of a 2-step PCR protocol.

**Statistical analysis and selection of primer pairs for absolute quantification.** Nonmetric multidimensional scaling (NMDS) with Bray-Curtis distance metrics was run to visualize the relative quantification results (i.e., $C_T$ values of targeted amoA-AOB genes – $C_T$ values of bacterial 16S rRNA genes) obtained from the screening run using 96.96 IFCs. An analysis of similarities (ANOSIM) test with Bray-Curtis distance metrics and 9,999 permutations was used to test the effect of different fertilizer levels on the relative abundances. Through a correlation analysis using the envfit function in R, 6 of the 78 primer pairs were selected for absolute quantification, as follows: amoA_AOB_p12, amoA_AOB_p31, amoA_AOB_p35, amoA_AOB_p42, amoA_AOB_p45, and amoA_AOB_p70. Statistical analysis was run with R version 4.0.3. NMDS, ANOSIM tests, and correlation analysis were performed using R package "vegan."

**Data availability.** For amoA-AOB primer design, 1,205 curated protein and corresponding nucleotide sequences and a file containing the mapping between each gene's nucleotide and protein accession identifier (ID) were obtained from the Fungene database (32) by filtering for a Hidden Markov Model (HMM) search score of >400 and HMM coverage over 70.2% amino acid similarity. To prioritize these gene targets for amoA-AOB function in soils, we used 1,550 publicly available soil metagenomes (Table S1) as reference metagenomes for primer design.

## SUPPLEMENTAL MATERIAL

Supplemental material is available online only.
**TEXT S1**, DOCX file, 0.02 MB.
**FIG S1**, DOCX file, 2.6 MB.
**FIG S2**, DOCX file, 2.3 MB.
**TABLE S1**, DOCX file, 0.1 MB.
**TABLE S2**, DOCX file, 0.02 MB.
**TABLE S3**, DOCX file, 0.02 MB.
**TABLE S4**, DOCX file, 0.02 MB.
**TABLE S5**, DOCX file, 0.02 MB.
**TABLE S6**, DOCX file, 0.02 MB.
**TABLE S7**, DOCX file, 0.01 MB.

## ACKNOWLEDGMENTS

This work was funded by the DOE Center for Advanced Bioenergy and Bioproducts Innovation (U.S. Department of Energy, Office of Science, Office of Biological and Environmental Research under award number DE-SC0018420).

Any opinions, findings, and conclusions or recommendations expressed in this publication are those of the author(s) and do not necessarily reflect the views of the U.S. Department of Energy.

This work was conducted under the MMPRNT project, funded by the DOE BER Office of Science award DE-SC0014108.

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
