## [Reviewer comments · mSystems]

MetaFunPrimer: an environment-specific, high-throughput primer design tool for improved quantification of target genes

Jia Liu, Paul Villanueva, Jin Choi, Santosh Gunturu, Yang Ouyang, Lisa Tiemann, James Cole, Kathryn Glanville, Steven Hall, Marshall McDaniel, Jaejin Lee, and Adina Howe

Corresponding Author(s): Adina Howe, Iowa State University

Review Timeline:

Submission Date:	February 23, 2021
Editorial Decision:	April 5, 2021
Revision Received:	July 23, 2021
Accepted:	August 23, 2021

Editor: Nick Bouskill

Reviewer(s): Disclosure of reviewer identity is with reference to reviewer comments included in decision letter(s). The following individuals involved in review of your submission have agreed to reveal their identity: Robert A Sanford (Reviewer #1); Sul Woo Jul (Reviewer #2)

Transaction Report:

DOI: <https://doi.org/10.1128/mSystems.00201-21>

April 5, 2021

Dr. Adina Howe
Iowa State University
Ames

Re: mSystems00201-21 (MetaFunPrimer: primer design for targeting genes observed in metagenomes)

Dear Dr. Adina Howe:

Thank you for your recent submission to mSystems. Your manuscript has now been reviewed by two experts in the field and both recommend revisions to the text prior to reconsideration. Below you will find the comments of the reviewers, and I would particularly encourage you to attend to reviewer #1's comments on the focus of the manuscript and also the need to improve the supplemental material significantly. I also agree with reviewer #2 when they ask for additional application of this approach.

To submit your modified manuscript, log onto the eJP submission site at <https://msystems.msubmit.net/cgi-bin/main.plex>. If you cannot remember your password, click the "Can't remember your password?" link and follow the instructions on the screen. Go to Author Tasks and click the appropriate manuscript title to begin the resubmission process. The information that you entered when you first submitted the paper will be displayed. Please update the information as necessary. Provide (1) point-by-point responses to the issues raised by the reviewers as file type "Response to Reviewers," not in your cover letter, and (2) a PDF file that indicates the changes from the original submission (by highlighting or underlining the changes) as file type "Marked Up Manuscript - For Review Only."

Due to the SARS-CoV-2 pandemic, our typical 60 day deadline for revisions will not be applied. I hope that you will be able to submit a revised manuscript soon, but want to reassure you that the journal will be flexible in terms of timing, particularly if experimental revisions are needed. When you are ready to resubmit, please know that our staff and Editors are working remotely and handling submissions without delay. If you do not wish to modify the manuscript and prefer to submit it to another journal, please notify me of your decision immediately so that the manuscript may be formally withdrawn from consideration by mSystems.

Sincerely,

Nick Bouskill

Editor, mSystems

Journals Department
Reviewer comments:

Reviewer #1 (Comments for the Author):

This manuscript describes the development of a computational tool to design qPCR primers for functional gene analysis. The authors use AOB amoA as a representative functional gene to demonstrate their software. The overall functionality and usefulness of the MetaFunPrimer program is well presented. Although there is much value in having tools like this for microbial ecology studies, there are some issues with how the example data is presented. I will detail these below.

1. There is nothing quantitative about the qPCR data presented yet the authors seem to sell their approach as a means to designing quantitative PCR primers. Admittedly one could use qPCR for each primer pair designed, however this is not what is presented in the example data. The authors used a Fluidigm qPCR system (one that I am familiar with) to test their 78 primer pairs against 30 soil samples from corn fields receiving different N-inputs. There are no standards run to calibrate the qPCR, although this would be difficult given 78 different primer sets. In addition some of the primers actually target the same related cluster of genes, so there would likely be cross amplification with some primers across target groups. Not to be entirely negative about the authors approach, I feel there is some benefit to doing what they did. It is just not appropriate to call it quantitative in the traditional way a reader would interpret this term. What the authors do is compare CT values across all DNA samples for all primer sets run. This creates a unique fingerprint for any sample that can be used to compare it to other samples. Also assuming that equivalent DNA masses are loaded into every sample slot, the relative CT values for primer pair qPCR reaction would give a relative difference in abundance between the samples for that particular target. This is semiquantitative at best.

2. I would not normally mention the quality of the Supplementary material as a major point of contention in a review, but I make an exception in this case. The supplemental tables, figures and methods are so poorly documented and organized that it almost made me not want to review this paper. If a reader wants to completely understand the example presented they have to access the Supplemental material. None of the material has a caption or title associated with it. Since it is supplemental there should be no reason that these Figures and Tables could have extra explanatory text affiliated with them. In addition the primer tables are confusing and could be better designed. For example the 78 primer pairs used should be listed with side by side F and R primers.

3. Although the authors do an excellent job demonstrating the successful design and application of primers targeting AOB in soils based on an assessment of 1550 metagenomes, I feel that things might get a little more complicated with other functional genes. It is clear from Figure S1 that the 60 gene clusters identified are for the most part not that distantly divergent and occur in mostly closely related taxa (i.e. narrow GC content range). This makes AOB amoA an ideal candidate for this approach because a relatively low number of primers could be designed (78) to cover >90% of the diversity in soils. Would the same be true if you looked at nirK, nirS, nrfA or nosZ? From my own experience with nrfA, which has 19 Clades delineated by 30% amino-acid sequence divergence, I would guess more than 78 primer pairs would be required to get >90% coverage. Even if we narrowed this down by ecosystem to seven or eight clades, I still think there would be more than 78 primer pairs. In addition the phylogenetic diversity of taxa containing nrfA genes is quite extraordinary, yielding GC contents ranging from <50% to 75%. This has an impact on PCR efficiency and potentially on how well the qPCR system will work. It would be nice if the authors would bring up some of these potential precautions or limits to how the method might be applied to other genes. Since we designed highly degenerate primers that theoretically have very good coverage for nrfA, it might be nice to convey why such degenerate primers might NOT be good for qPCR and that a multiplexed qPCR system approach is a better way to at least get some semiquantitative information from this approach.

4. I realize most of my comments are addressing the example provided and not the primary product of the program described. The reason I did this is because this example is critical to demonstrating how one might benefit from the use of this software. Therefore it is important to more completely document this example. I note that the authors state that they used the "Standard Methods" for qPCR defined by the Fluidigm system. I guarantee that Standard protocol for qPCR provided by Fluidigm did not have functional genes from soil in mind. For example, we did some work with the same platform and were required to preamplify the genes in the DNA samples before putting them on the microarray. Did you have to do this? These details must be included in this manuscript.

5. Ln 158-159: The authors state that there are 28 primer pairs generated in EcoFunPrimer and refer us to Table S3. Table S3 has more than 28 primers, so it is unclear what the authors are referring to here.

6. Ln 160: Seems to refer to the same 28 "degenerate" primer pairs apparently shown in Table S3, however there are no degenerate primers shown in this confusing table.

7. Table S4: Modify the table to show a list of 78 primer pairs with the forward and reverse side by side.

8. Table S5: No caption is provided and the entire table is very confusing. I have no idea what the Primer Set ID is or what the other primer name designation come from for "this study". Since the authors refer to this as a comparison with previously designed primers, it is critical that they provide substantially more explanative documentation with the table.

9. Table 2: A general question hit me when I saw this table. If there are 26 MetaFunPrimer pairs designed for Cluster 4, do they potentially have different predicted PCR efficiencies to the 320 potential targets, thus potentially compromising interpretation of qPCR data?

10. Ln 186-188: I doubt the authors meant that the two primer pairs have not yet been characterized. I suspect that metagenomic derived amoA gene targets are present in organisms that are not yet characterized. Also, how do the two primer pairs mentioned explain the variation

shown in Figure 4? More should be included in the text and in the Figure caption. Also typically ANOSIM has an R value associated with it. This should be included in Figure 4.

11. Ln 207-208: Not sure why "novel" is used here. These are simply new primers. Also I'm not sure how you are showing improved quantification. In case I missed something, please elaborate.

Reviewer #2 (Comments for the Author):

Lui et al., describes the tool for designing multiplex qPCR primer sets based on metagenomic data. MetaFunPrimer is basically based on collective sequences of the target gene from multiple sources: metagenome data and previously deposited reference sequences Database. It is suitable to design multiplex primers set. Overall pipeline is acceptable though some subjective parameters such as 96% aa similarity they used should be discussed thoroughly why the author chose these values. minor comments:

1. There is a certain portion of the false-positive signal in Ct values by qPCR. Validation of these primers set- amount of false-positive may be required. Amplicon sequencing by MiSeq with comparing sequences on some soil sample to validate ratio of false-positive in qPCR primers.

2. Even though authors extensively analyzed the amoA-AOB case, readers may still be perplexing which genes are applicable to MetaFunPrimer pipeline. Please describe or explain criteria when they can use this pipeline. i.e. How many reference gene are needed?

Or like microcystin-producing genes in hundreds of lake water samples you mentioned, describe more examples using MetaFunPrimer pipeline.

The line numbers in the reviewers' comments refer to the original manuscript, whereas those in our **Response** refer to the revised manuscript. Our Responses are in **blue** text.

Reviewer #1 (Comments for the Author):

This manuscript describes the development of a computational tool to design qPCR primers for functional gene analysis. The authors use AOB amoA as a representative functional gene to demonstrate their software. The overall functionality and usefulness of the MetaFunPrimer program is well presented. Although there is much value in having tools like this for microbial ecology studies, there are some issues with how the example data is presented. I will detail these below.

Response: We really appreciate your thoughtful comments and suggestions. We admit that our presentation of example data was not very efficient. We hope that our efforts to improve the manuscript have clarified your concerns.

1. There is nothing quantitative about the qPCR data presented yet the authors seem to sell their approach as a means to designing quantitative PCR primers. Admittedly one could use qPCR for each primer pair designed, however this is not what is presented in the example data. The authors used a Fluidigm qPCR system (one that I am familiar with) to test their 78 primer pairs against 30 soil samples from corn fields receiving different N-inputs. There are no standards run to calibrate the qPCR, although this would be difficult given 78 different primer sets. In addition some of the primers actually target the same related cluster of genes, so there would likely be cross amplification with some primers across target groups. Not to be entirely negative about the authors approach, I feel there is some benefit to doing what they did. It is just not appropriate to call it quantitative in the traditional way a reader would interpret this term. What the authors do is compare CT values across all DNA samples for all primer sets run. This creates a unique fingerprint for any sample that can be used to compare it to other samples. Also assuming that equivalent DNA masses are loaded into every sample slot, the relative CT values for primer pair qPCR reaction would give a relative difference in abundance between the samples for that particular target. This is semiquantitative at best.

Response: We do understand why our previous manuscript may be considered a 'semiquantitative' approach. We have added text to clarify the quantitative power of our approach. Additionally, we have used standards to provide absolute quantification in our example with a new set of experiments (Line 209-225, Line 329-387, Fig. 5, Fig. S2, Table S6). We have clarification on a few points detailed below.

- 1) We feel that we must first acknowledge that the present paper is mainly about computational tool, "MetaFunPrimer", which can produce the least number of primer pairs to cover given sequences. A natural and important complement to any computational tool is experimental validation, though this evidence does not always accompany computational tool publications. Initially, in the previous version of the manuscript, we were hesitant to showcase this tool with too specific a platform (e.g., high throughput qPCR) because we feel like it may diminish its perception of its broad

applicability (e.g., for qPCR which has more democratized access). However, we considered the reviewer's suggestion and felt like it does strengthen this manuscript to include experimental validation of our primers with HT-qPCR. We have also added text to note that the usage of MetaFunPrimer is not limited to HT-qPCR (Line 276-280).

- 2) To demonstrate the quantitative nature that can be achieved with MetaFunPrimer and HT-qPCR, we chose 6 primer pairs and performed absolute quantification using standard DNA samples. The selection of these primers was based on first a screen of 78 primers, as users may also implement for selecting primers for absolute quantification. We have also added text to outline strategies for absolute and relative quantification within this example (Line 209-225, Line 329-387, Fig. 5, Fig. S2, Table S6).
- 3) As the reviewer mentioned, there generally can exist cross-amplification between primer pairs. Our bioinformatic pipeline minimizes cross-amplification. EcoFunPrimer, which is embedded in MetaFunPrimer, outputs multiple primer pairs for the same gene target in some cases. Next, MetaFunPrimer has a function to optimize and select the minimal set of primer pairs that can exclusively target the maximal diversity of functional genes of interest (Line 191-193). Experimentally, it is possible that unintentional cross amplification may occur during the PCR process. The likelihood of this occurring is also related to the experimental conditions chosen (e.g., HT-qPCR vs standard qPCR, the number of probes, etc.). However, since this is part of the experimental process (e.g., qPCR optimization), we decided that this was outside the scope of this manuscript. Our addition of the experimental validation of the primers designed helps to demonstrate the value of the software, but we fully acknowledge that experimental optimization may be needed (Line 260-262, Line 265-266, Line 273-274, Line 300-301).

2. I would not normally mention the quality of the Supplementary material as a major point of contention in a review, but I make an exception in this case. The supplemental tables, figures and methods are so poorly documented and organized that it almost made me not want to review this paper. If a reader wants to completely understand the example presented they have to access the Supplemental material. None of the material has a caption or title associated with it. Since it is supplemental there should be no reason that these Figures and Tables could have extra explanatory text affiliated with them. In addition the primer tables are confusing and could be better designed. For example the 78 primer pairs used should be listed with side by side F and R primers.

Response: Inexcusably, our supplementary tables and figures were not well-organized in the previous manuscript. We apologize and have learned our lesson. We thank the reviewers for their patience and mentorship. We have revised all the tables and figures. The table containing primer information has been modified as suggested by the reviewer (Table S4).

3. Although the authors do an excellent job demonstrating the successful design and application of primers targeting AOB in soils based on an assessment of 1550 metagenomes, I feel that things might get a little more complicated with other functional genes. It is clear from Figure S1 that the 60 gene clusters identified are for the most part not that distantly divergent and occur in mostly closely related taxa (i.e. narrow GC content range). This makes AOB amoA an ideal candidate for this approach because a relatively low number of primers could be designed (78) to cover >90% of the diversity in soils. Would the same be true if you looked

at nirK, nirS, nrfA or nosZ? From my own experience with nrfA, which has 19 Clades delineated by 30% amino-acid sequence divergence, I would guess more than 78 primer pairs would be required to get >90% coverage. Even if we narrowed this down by ecosystem to seven or eight clades, I still think there would be more than 78 primer pairs. In addition the phylogenetic diversity of taxa containing nrfA genes is quite extraordinary, yielding GC contents ranging from <50% to 75%. This has an impact on PCR efficiency and potentially on how well the qPCR system will work. It would be nice if the authors would bring up some of these potential precautions or limits to how the method might be applied to other genes. Since we designed highly degenerate primers that theoretically have very good coverage for nrfA, it might be nice to convey why such degenerate primers might NOT be good for qPCR and that a multiplexed qPCR system approach is a better way to at least get some semiquantitative information from this approach.

Response: Thank you for your comments. We acknowledge that depending on the functional gene targets of interest, challenges for primer design will vary. This study was based on *amoA*-AOB because of its impacts associated with nitrogen turnover in managed agroecosystems, for which we developed this tool. The reviewer is correct, in our experience, if primer pairs are designed for other nitrogen cycle genes, some of the genes (e.g. *narG*, *napA*, etc.) are too diverse to design a reasonable number of primer pairs to cover even >50% of reference sequences. We mentioned it in Line 262-265. Following the reviewer's suggestion, we have added to text to discuss applications of these tools and user choices to accommodate varying needs (Line 163-165, Line 260-262, Line 265-266). We have also mentioned that universal primers or degenerate primers can be limited due to low resolution in some cases and may lead to the loss of the ability to identify specific bacterial species or strains (Line 90-96). Additionally, we have highlighted that MetaFunPrimer can also generate degenerate primers (Line 184-186, Line 318-320). An option is available for users to choose how much degeneracy can be allowed in a primer. In the text, we focused mainly on the features of MetaFunPrimer, with some comments on potential application to available platforms. Thus, we wrote mainly about the capabilities and advantages/disadvantages of degenerate primers but specifically did not comment on specific platforms.

4. I realize most of my comments are addressing the example provided and not the primary product of the program described. The reason I did this is because this example is critical to demonstrating how one might benefit from the use of this software. Therefore it is important to more completely document this example. I note that the authors state that they used the "Standard Methods" for qPCR defined by the Fluidigm system. I guarantee that Standard protocol for qPCR provided by Fluidigm did not have functional genes from soil in mind. For example, we did some work with the same platform and were required to preamplify the genes in the DNA samples before putting them on the microarray. Did you have to do this? These details must be included in this manuscript.

Response: We hope that we have struck a balance between describing MetaFunPrimer and including an experimental validation of its application. We agree that the methods we used need to be more thoroughly documented. Preamplification can be a reasonable way to deal with environmental samples, but we tested our primers and soil samples in various ways not to

include the preamplification step. Our concern was that this step could also be another alteration to the samples. We found that a three-step PCR process for 10-fold diluted soil DNA samples had good amplifications. Please see the results in Fig. 4, Fig. 5, Fig. S2, and Table S7. The details of these methods have also been added to the manuscript (Line 329-387, Material S1).

5. Ln 158-159: The authors state that there are 28 primer pairs generated in EcoFunPrimer and refer us to Table S3. Table S3 has more than 28 primers, so it is unclear what the authors are referring to here.

Response: Table S3 has been revised to clearly show the 20 degenerate and 8 non-degenerate primer pairs. The RDP EcoFunPrimer (<https://github.com/rdpstaff/EcoFunPrimer>) is the main embedded primer design tool of MetaFunPrimer, and MetaFunPrimer is an expanded version of EcoFunPrimer. EcoFunPrimer can generate both degenerate and non-degenerate primers by setting the extent of degeneracy option.

6. Ln 160: Seems to refer to the same 28 "degenerate" primer pairs apparently shown in Table S3, however there are no degenerate primers shown in this confusing table.

Response: We apologize that our previous table confused the reviewer. Table S3 has been revised to clearly show the 20 degenerate primer pairs and their all possible combinations, separated from 8 non-degenerate primer pairs.

7. Table S4: Modify the table to show a list of 78 primer pairs with the forward and reverse side by side.

Response: Revised as suggested.

8. Table S5: No caption is provided and the entire table is very confusing. I have no idea what the Primer Set ID is or what the other primer name designation come from for "this study". Since the authors refer to this as a comparison with previously designed primers, it is critical that they provide substantially more explanative documentation with the table.

Response: Table S5 has been revised. The confusing IDs were for our internal identification for the computational analysis. We simplified the table and deleted the confusing IDs, and we apologize.

9. Table 2: A general question hit me when I saw this table. If there are 26 MetaFunPrimer pairs designed for Cluster 4, do they potentially have different predicted PCR efficiencies to the 320 potential targets, thus potentially compromising interpretation of qPCR data?

Response: GC contents of the 320 target genes in Cluster 4 range from 57.2% to 61.6%. The melting temperature for the primers were set to 60°C. Thus, we concluded there would not much difference in PCR efficiencies. Please see the figure below. Additionally, to ensure designed primer pairs to have similar PCR efficiencies, melting temperature and GC contents can be defined by users.

10. Ln 186-188: I doubt the authors meant that the two primer pairs have not yet been characterized. I suspect that metagenomic derived amoA gene targets are present in organisms that are not yet characterized. Also, how do the two primer pairs mentioned explain the variation shown in Figure 4? More should be included in the text and in the Figure caption. Also typically ANOSIM has an R value associated with it. This should be included in Figure 4.

Response: We performed a new set of experiments and added detailed information as suggested (Fig. 4, Fig. 5, Fig. S2, Table S6, Line 209-225, Line 329-387).

11. Ln 207-208: Not sure why "novel" is used here. These are simply new primers. Also I'm not sure how you are showing improved quantification. In case I missed something, please elaborate.

Response: We have modified this text according to this comment. We hope a new set of experiments consisting of two steps can demonstrate the advancement of MetaFunPrimer tool in quantifying target genes at higher resolution.

Reviewer #2 (Comments for the Author):

Lui et al., describes the tool for designing multiplex qPCR primer sets based on metagenomic data. MetaFunPrimer is basically based on collective sequences of the target gene from multiple sources: metagenome data and previously deposited reference sequences Database. It is suitable to design multiplex primers set. Overall pipeline is acceptable though some subjective parameters such as 96% aa similarity they used should be discussed thoroughly why the author chose these values.

Response: We really appreciate your thoughtful comments and suggestions. The reviewer made a valid point. Without detailed information, setting such a threshold can be seen as subjective. Detailed information regarding the calculation can be found at https://github.com/jialiu232/MetaFunPrimer_paper_info.git. The calculation process is already embedded in MetaFunPrimer tool and it is mentioned in the revised manuscript (Line 298-300).

Minor comments:

1. There is a certain portion of the false-positive signal in Ct values by qPCR. Validation of these primers set- amount of false-positive may be required. Amplicon sequencing by MiSeq with comparing sequences on some soil sample to validate ratio of false-positive in qPCR primers.

Response: Thank you for the suggestions. We do agree that false positives should be excluded from the analysis. Instead of performing amplicon sequencing analysis, we believe that the standard curves generated by selected primer sets can provide us the guideline on false positives (Fig. S2). For the 96.96 IFCs that did not include DNA standards, we excluded any amplifications which Ct values were over 23 from further analysis, based on the multiple test runs to identify limit of detection. (Line 371-372) Setting a threshold cycle cutoff is an acceptable method for high throughput qPCR approaches as shown in literature. (For instance, Stedfeld et al., 2018, *FEMS Microbiology Ecology*, <https://doi.org/10.1093/femsec/fiy130>).

2. Even though authors extensively analyzed the amoA-AOB case, readers may still be perplexing which genes are applicable to MetaFunPrimer pipeline. Please describe or explain criteria when they can use this pipeline. i.e. How many reference gene are needed? Or like microcystin-producing genes in hundreds of lake water samples you mentioned, describe more examples using MetaFunPrimer pipeline.

Response: Fundamentally, MetaFunPrimer is an extension to EcoFunPrimer. EcoFunPrimer has been applied to various genes including antibiotic resistance genes (Stedfeld et al., 2018), genes related to RDX biodegradation (Collier et al., 2019), microcystin-producing genes (Lee et al., 2020), and foodborne pathogens (Williams and Hashsham, 2019).

This paper introduces a tool that can produce a single to hundreds of primer pairs to cover the given reference genes. Parameters, including melting temperature and GC content, can be defined by the user (Table S6). This tool can also be used to for other functional genes, as long as sequences are available. The applicability, the appropriate options, and the number of environment-specific metagenomes may be explored by users and is designed for flexibility to

varying user questions and needs. Metagenomes are also an optional input parameter and users can input relevant genomes instead of metagenomes (Line 273-274).

August 23, 2021

Dr. Adina Howe
Iowa State University
Ames

Re: mSystems00201-21R1 (MetaFunPrimer: an environment-specific, high-throughput primer design tool for improved quantification of target genes)

Dear Dr. Adina Howe:

Your manuscript has been accepted, and I am forwarding it to the ASM Journals Department for publication. For your reference, ASM Journals' address is given below. Before it can be scheduled for publication, your manuscript will be checked by the mSystems senior production editor, Ellie Ghatineh, to make sure that all elements meet the technical requirements for publication. She will contact you if anything needs to be revised before copyediting and production can begin. Otherwise, you will be notified when your proofs are ready to be viewed.

As an open-access publication, mSystems receives no financial support from paid subscriptions and depends on authors' prompt payment of publication fees as soon as their articles are accepted. =

Publication Fees:

We recognize that the video files can become quite large, and so to avoid quality loss ASM suggests sending the video file via <https://www.wetransfer.com/>. When you have a final version of the video and the still ready to share, please send it to Ellie Ghatineh at eghatineh@asmusa.org.

Sincerely,

Nick Bouskill
Editor, mSystems

Journals Department
Table S7: Accept
Table S3: Accept
Fig.S1: Accept
Material S1: Accept
Table S4: Accept
Fig.S2: Accept
Table S5: Accept
Table S2: Accept
Table S6: Accept
Table S1: Accept